



# Deep learning rainfall-runoff predictions of extreme events

Jonathan M. Frame[1,2], Frederik Kratzert[3], Daniel Klotz[3], Martin Gauch[3], Guy Shalev[4], Oren Gilon[4], Logan M. Qualls[2], Hoshin V. Gupta[5], and Grey S. Nearing[6,7]

[1]National Water Center, National Oceanic and Atmospheric Administration, Tuscaloosa, AL, United States
[2]University of Alabama, Tuscaloosa, AL, United States
[3]LIT AI Lab & Institute for Machine Learning, Johannes Kepler University, Linz, Austria
[4]Google Research, Tel Aviv, Israel
[5]The University of Arizona, Tucson, AZ, United States
[6]Google Research, Mountain View, CA, United States
[7]University of California Davis, Department of Land, Air & Water Resources, Davis, CA, United States

**Correspondence:** jmframe@crimson.ua.edu

**Abstract.** The most accurate rainfall-runoff predictions are currently based on deep learning. There is a concern among hydrologists that data-driven models based on deep learning may not be reliable in extrapolation or for predicting extreme events. This study tests that hypothesis using Long Short-Term Memory networks (LSTMs) and an LSTM variant that is architecturally constrained to conserve mass. The LSTM (and the mass-conserving LSTM variant) remained relatively accurate in predict-
ing extreme (high return-period) events compared to both a conceptual model (the Sacramento Model) and a process-based model (US National Water Model), even when extreme events were not included in the training period. Adding mass balance constraints to the data-driven model (LSTM) reduced model skill during extreme events.

## 1 Introduction

Deep learning (DL) provides the most accurate rainfall-runoff simulations available from the hydrological sciences community
(Kratzert et al., 2019b, a). This type of finding is not new – Todini (2007) noted more than a decade ago, in his review of the history of hydrological modeling, that *"physical process-oriented modellers have no confidence in the capabilities of data-driven models' outputs with their heavy dependence on training sets, while the more system engineering-oriented modellers claim that data-driven models produce better forecasts than complex physically-based models."* Echoing this sentiment about the perceived reliability of data-driven models, Sellars (2018) reported in their summary of a workshop on 'Big Data and the
Earth Sciences' that *"[m]any participants who have worked in modeling physical-based systems continue to raise caution about the lack of physical understanding of ML methods that rely on data-driven approaches."*

The idea that hydrological models based on physical understanding might be more reliable than machine learning (ML) based models in out-of-sample conditions was drawn from early experiments on shallow neural networks (e.g., Cameron et al., 2002; Gaume and Gosset, 2003). However, although this idea is still frequently cited (e.g., quotations above; Herath et al.,
2020; Reichstein et al., 2019; Rasp et al., 2018), it has not been tested in the context of modern DL models, which are able to generalize complex hydrological relationships across space and time (Nearing et al., 2020b). Further, there is some evidence





that this hypothesis might not be true. For example, Kratzert et al. (2019a) showed that DL can generalize to *ungauged* basins with better overall skill than calibrated conceptual models in *gauged* basins. Kratzert et al. (2019b) used a slightly modified version of a Long Short-Term Memory Network (LSTM) to show how the model learns to transfer information between basins.

Similarly, Nearing et al. (2019) showed how an LSTM-based model learns *dynamic* basin similarity under changing climate, so that when the climate in a particular basin shifts (e.g., becomes wetter or drier), the model learns to adapt hydrological behavior based on different climatological neighbors. Further, because DL is currently the state-of-the-art for rainfall-runoff prediction, it is important to understand its potential limits.

The primary objective of this study is to test the hypothesis that data-driven models lose reliability in extreme events more

than models based on process-understanding. We focus specifically on high return period (low probability) streamflow events, and compare four models: a standard deep learning model, a physics-informed deep learning model, a conceptual rainfall-runoff model, and a process-based hydrological model.

## 2   Methods

### 2.1   Data

The hydrological sciences community lacks community-wide standardized procedures for model benchmarking, which severely limits the effectiveness of new model development and deployment efforts (Nearing et al., 2020b). In previous studies, we used open community data sets and consistent training/test procedures that allow for results to be directly comparable between studies – we continue that practice here to the extent possible.

Specifically, we used the Catchment Attributes and Meteorological Large Sample (CAMELS) data set curated by the US

National Center for Atmospheric Research (NCAR) (Newman et al., 2015; Addor et al., 2017). The CAMELS data set consists of daily meteorological and discharge data from 671 catchments in CONUS ranging in size from 4 $km^2$ to 25,000 $km^2$ that have largely natural flows and long streamflow gauge records (1980-2008). We used 498 of 671 CAMELS catchments – these were included in the basins that were used for model benchmarking by Newman et al. (2017), who removed basins with (i) large discrepancies between different methods of calculating catchment area, and (ii) areas larger than 2,000 $km^2$.

CAMELS includes daily discharge data from the USGS Water Information System, which are used as training and evaluation target data. CAMELS also includes several daily meteorological forcing data sets (Daymet, NLDAS, Maurer). We used NLDAS for this project because we benchmarked against the National Water Model retrospective (will be introduced in detail in 2.3.2), which also uses NLDAS. CAMELS also includes several static catchment attributes related to soils, climate, vegetation, topography, and geology (Addor et al., 2017) that are used as input features. We used the same input features (meteorological

forcings and static catchment attributes) that were listed in Table 1 by Kratzert et al. (2019b).





## 2.2 Return Period Calculations

The return periods of peak annual flows provide a basis for categorizing target data in a hydrologically meaningful way. This results in a metric that is consistent while maintaining diversity across basins – e.g., a similar flow volume may be 'extreme' in one basin but not in another. Splitting model training and test periods by different return periods allows us to assess model performance on both rare and effectively unobserved events.

For return period calculations we followed guidelines in the U.S. Interagency Committee on Water Data Bulletin 17b (IACWD, 1982). The procedure is to fit all available annual peak flows (log transformed) for each basin to a Pearson Type III distribution using the method of moments:

$$f(y; \tau, \alpha, \beta) = \frac{(\frac{y-\tau}{\beta})^{\alpha-1} exp(-\frac{y-\tau}{\beta})}{|\beta| \Gamma(\alpha)}, \tag{1}$$

with $\frac{y-\tau}{\beta} > 0$ and distribution parameters $\tau$, $\alpha$, and $\beta$, where $\tau$ is the location parameter, $\alpha$ is the shape parameter, $\beta$ is the scale parameter, and $\Gamma(\alpha)$ is the gamma function.

To calculate the return periods, we used annual peak flow observations taken directly from the USGS National Water Information System (WIS), instead of from the CAMELS data, because the Bulletin 17b guidelines require annual peak flows whereas CAMELS provides only daily averaged flows. The Bulletin 17b (IACWD, 1982) guidelines require using all available data, which for peak flows ranges from 26 to 116 years. After fitting the return period distributions for each basin, we classified each water year of the CAMELS data from each basin (each basin-year of data) according to the return period of its observed peak annual discharge.

This return-period analysis does not account for nonstationarity – i.e., the return period of a given magnitude of event in a given basin could change due to changing climate or changing land use. There is currently no agreed upon method to account for nonstationarity when determining flood flow frequencies, so it would be difficult to incorporate this in our return period calculations. However, for the purpose of this paper (testing whether the LSTM is reliable in extreme events) this is not an issue because stationary return period calculations directly test predictability on large events that are out-of-sample *relative to the training period*, which for practical purposes can represent potential nonstationarity.

## 2.3 Models

### 2.3.1 ML Models & Training

We test two ML models: a pure LSTM and a physics-informed LSTM that is architecturally constrained to conserve mass – we call this a Mass-Conserving LSTM (MC-LSTM; Hoedt et al., 2021). These models are described in detail in Appendices A and B.

Daily meteorological forcing data and static catchment attributes data were used as inputs features for the LSTM and MC-LSTM, and daily streamflow records were used as training targets with a normalized squared-error loss function that does not





depend on basin-specific mean discharge (i.e., large and/or wet basins are not over-weighted in the loss function):

$$\text{NSE*} = \frac{1}{B} \sum_{b=1}^{B} \sum_{n=1}^{N} \frac{(\widehat{y}_n - y_n)^2}{(s(b) + \epsilon)^2},$$ (2)

where $B$ is the number of basins, $N$ is the number of samples (days) per basin $B$, $\widehat{y}_n$ is the prediction for sample $n$ ($1 \le n \le N$), $y_n$ is the corresponding observation, and $s(b)$ is the standard deviation of the discharge in basin $b$ ($1 \le b \le B$), calculated from the training period (see, Kratzert et al., 2019b).

We trained both the standard LSTM and the MC-LSTM using the same training and test procedures outlined by Kratzert et al. (2019b). Both models were trained for 30 epochs using sequence-to-one prediction to allow for randomized, small minibatches. We used a minibatch size of 256 and, due to sequence-to-one training, each minibatch contained (randomly selected) samples from multiple basins. The standard LSTM had 128 cell states and a 365-day sequence length. Input and target features for the standard LSTM were pre-normalized by removing bias and scaling by variance. For the MC-LSTM the inputs were split between auxiliary, which were pre-normalized, and the mass input (in our case precipitation), which was not pre-normalized. Gradients were clipped to a global norm (per minibatch) of 1. Heteroscedastic noise was added to training targets (resampled at each minibatch) with standard deviation of 0.005 times the value of each target datum. We used an Adam optimizer with a fixed learning rate schedule; the initial learning rate of 1e-3 was decreased to 5e-4 after 10 epochs and 1e-4 after 25 epochs. Biases of the LSTM forget gate were initialized to 3 so that gradient signals persisted through the sequence from early epochs.

The MC-LSTM used the same hyperparameters as the LSTM except that it used only 64 cell states, which was found to perform better for this model (see, Hoedt et al., 2021). Note that the memory states in an MC-LSTM are fundamentally different than those of the LSTM due to the fact that they are physical states with physical units instead of purely information states.

All ML models were trained on data from the CAMELS catchments simultaneously. We used three different train and test periods:

1. The first train/test period split was the same split used in previous studies (Kratzert et al., 2019b, 2021; Hoedt et al., 2021). In this case, the training period included nine water years from October 1, 1999 through September 30, 2008, and the test period included ten water years 1990-1999 (i.e., from October 1, 1989 through September 30, 1999). This train/test split was used *only* to ensure that the models trained here achieved similar performance compared with previous studies.

2. The second train/test period split used a test period that aligns with the availability of benchmark data from the US National Water Model (see Section 2.3.2). The train period included water years 1981-1995, and the test period included water years 1996-2014 (i.e., from October 1, 1995 through September 30, 2014). This was the same training period used by Newman et al. (2017) and Kratzert et al. (2019a), but with an extended test period. This train/test split was used because the NWM-Rv2 data record is not long enough to accommodate the train/test split used by previous studies (item above in this list).

3. The third train/test period split used all water years in the CAMELS data set with five-year or lower return period peak flow for training, while the test period included water years with greater than five-year return period peak flow in the





period 1996-2014 (to be comparable with the test period in the item above). This is to test whether the data-driven models
can extrapolate to extreme events that are not included in the training data. Return period calculations are described in
Section 2.2. To account for the 365-day sequence length for sequence-to-one prediction, we separated all train and test
years in each basin by at least one year (i.e., we removed years with high return periods, and their preceding years, from
the training set). A file containing the train/test year splits for each CAMELS basin based on return periods is available
in the GitHub repository linked in the Code and Data Accessibility statement.

### 2.3.2 Benchmark Models & Calibration

The conceptual model that we used as a benchmark was the Sacramento Soil Moisture Accounting model (SAC-SMA) with
SNOW-17 and a unit hydrograph routing function. This same model was used by (Newman et al., 2017) to provide standard-
ized model benchmarking data as part of the CAMELS data set, however we re-calibrated SAC-SMA to be consistent with
our training/test splits that are based on return periods. We used the Python-based SAC-SMA code and calibration package
developed by (Nearing et al., 2020a), which uses the SpotPy calibration library (Houska et al., 2019). SAC-SMA was calibrated
separately at each of the 531 CAMELS basins using the three train/test splits outlined in Section 2.3.1.

The process-based model that we used as a benchmark was the NOAA National Water Model (NWM) retrospective run
version 2 (NWM-Rv2). The NWM is based on WRF-Hydro (Salas et al., 2018), which is a process-based model that includes
Noah-MP (Niu et al., 2011) as a land surface component, kinematic wave overland flow, and Muskingum-Cunge channel
routing. NWM-Rv2 was previously used as a benchmark for LSTM simulations in CAMELS by Kratzert et al. (2019a), Gauch
et al. (2021) and Frame et al. (2020). Public data from NWM-Rv2 is hourly and CONUS-wide – we pulled hourly flow estimates
from the USGS gauges in the CAMELS data set and averaged these hourly data to daily over the time period October 1, 1980
through September 30, 2014. As a point of comparison, Gauch et al. (2021) compared hourly and daily LSTM predictions
against the NWM-Rv2 and found that the NWM-Rv2 was significantly more accurate at the daily timescale than at the hourly
timescale, whereas the LSTM did not lose accuracy at the hourly timescale vs. the daily timescale. All experiments in the
present study were done at the daily timescale.

The NWM-Rv2 was calibrated by NOAA personnel on about 1400 basins with NLDAS forcing data on water years 2009-
2013. Part of our experiment and analysis includes data-driven models trained on irregular years, specifically with water years
that include peak flow annual return period less than 5 years, and the calibration of the conceptual model (SAC-SMA) was also
done on these years. Without the ability to re-calibrate the NWM-Rv2 on the same time period as the LSTM, MC-LSTM and
SAC-SMA, we cannot directly compare the performance of the NWM-Rv2 with the other models. This model still provides a
useful benchmark for the data-driven models, even if it does have a slight advantage over the other models due to the calibration
procedure.

### 2.3.3 Performance Metrics and Assessment

We used the same set of performance metrics that were used in previous CAMELS studies (Kratzert et al., 2019b, a, 2021;
Gauch et al., 2021; Klotz et al., 2021). A full list of these metrics is given in Table 1. Each of the metrics was calculated





**Table 1.** Overview of evaluation metrics. The notation of the original publications is kept.

| Metric | Description | Reference/Equation |
|---|---|---|
| NSE[i] | Nash-Sutcliff efficiency | Eq. 3 in Nash and Sutcliffe (1970) |
| KGE[ii] | Kling-Gupta efficiency | Eq. 9 in Gupta et al. (2009) |
| Pearson-r | Pearson correlation between observed and simulated flow | |
| $\alpha$-NSE[iii] | Ratio of standard deviations of observed and simulated flow | From Eq. 4 in Gupta et al. (2009) |
| $\beta$-NSE[iv] | Ratio of the means of observed and simulated flow | From Eq. 10 in Gupta et al. (2009) |
| FHV[v] | Top 2% peak flow bias | Eq. A3 in Yilmaz et al. (2008) |
| FLV[vi] | Bottom 30% low flow bias | Eq. A4 in Yilmaz et al. (2008) |
| FMS[vii] | Bias of the slope of the flow duration curve between the 20% and 80% percentile | Eq. A2 Yilmaz et al. (2008) |
| Peak-Timing[viii] | Mean peak time lag (in days) between observed and simulated peaks | Appendix B in Kratzert et al. (2021)[ix] |
| Abs. error peak Q | Absolute percent error of peak flow | $\left(\frac{\|Q_{obs}-Q_{sim}\|}{Q_{obs}}\right)$. |

[i]: *Nash-Sutcliffe efficiency:* $(-\infty, 1]$, *values closer to one are desirable.*

[ii]: *Kling-Gupta efficiency:* $(-\infty, 1]$, *values closer to one are desirable.*

[iii]: *$\alpha$-NSE decomposition:* $(0, \infty)$, *values close to one are desirable.*

[iv]: *$\beta$-NSE decomposition:* $(-\infty, \infty)$, *values close to zero are desirable.*

[v]: *Top 2 % peak flow bias:* $(-\infty, \infty)$, *values close to zero are desirable.*

[vi]: *Bias of FDC midsegment slope:* $(-\infty, \infty)$, *values close to zero are desirable.*

[vii]: *30 % low flow bias:* $(-\infty, \infty)$, *values close to zero are desirable.*

[viii]: *Peak-timing:* $(-\infty, \infty)$, *values close to zero are desirable.*

[ix]: *This is a slightly different metric than described by Kratzert et al. (2021) in that we report the mean **absolute** peak time lag.*

for each basin separately on the whole test period for each of the training/test splits described in Section 2.3.1 except for the return-period based training/test split. In the former case (contiguous training/test periods) our objective is to maintain continuity with previous studies that report statistics calculated over entire test periods. In the latter case (return-period based
training/test splits) our objective is to report statistics separately for different return periods, and it is therefore necessary to calculate separate metrics for each water year and each basin in the test period. The last metric outlined in Table 1, the absolute percent bias of peak flow only for the largest streamflow event in each water year, lets us assess the ability to extrapolate to high-flow events. The metric was calculated separately for each annual peak flow event in all three training/test splits.

## 3 Results

### 3.1 Benchmarking Whole Hydrographs

Table 2 provides performance metrics for all models (Section 2.3.2) on the three test periods (Section 2.3.1). Appendix C provides a breakdown of the metrics in Table 2 by annual return period.

The first test period (1989-1999) is the same period used by previous studies, which allows us to confirm that the DL-based models (LSTM and MC-LSTM) trained for this project perform as expected relative to prior work. The performance of these





models (according to the metrics) are broadly equivalent to those reported for single models (not ensembles) by Kratzert et al. (2019b) (LSTM) and Hoedt et al. (2021) (MC-LSTM).

The second test period (1995-2014) allows us to benchmark against the NWM-Rv2, which does not provide data prior to 1995. Most of these scores are broadly equivalent to the metrics for the same models reported for the test period 1989-1999, with the exception of the FHV (high flow bias), FLV (low flow bias), add FMS (flow duration curve bias). These metrics depend

heavily on the observed flow characteristics during a particular test period and, because they are less stable, are somewhat less useful in terms of drawing general conclusions. We report them here primarily for continuity with previous studies (Kratzert et al., 2019b, a, 2021; Frame et al., 2020; Nearing et al., 2020a; Klotz et al., 2021; Gauch et al., 2021), and because one of the objectives of this paper (Section 2.2) is to expand on the high flow (FHV) analysis by benchmarking on annual peak flows.

The third test period (based on return periods) allows us to benchmark only on water years that contain streamflow events that

are larger (per basin) than anything seen in the training data (<= 5-year return periods in training and > 5-year return periods in testing). Model performances generally improve overall in this period according to the three correlation-based metrics (NSE, KGE, Pearson-r), but degrade according to the variance-based metric (alpha-NSE). This is expected due to the nature of the metrics themselves – hydrology models generally exhibit higher correlation with observations under wet conditions, simply due to higher variability. However, the data-driven models remained better than both benchmark models against all four of

these metrics, and while the bias metric (beta-NSE) was less consistent across test periods, the data-driven models had less overall bias than both benchmark models in the return-period test period.

The results in Table 2 indicate broadly similar performance between the LSTM and MC-LSTM across most metrics in the two nominal (i.e., unbiased) test periods. However, there were small differences. The MC-LSTM generally performed slightly worse according to most metrics and test periods. The cross-comparison was mixed according to the timing-based metric

(Peak-Timing). Notably, differences between the two ML-based models were small compared to the differences between these models and the conceptual (SAC-SMA) and process-based (NWM-Rv2) models, which both performed substantively worse across all metrics except FLV and FMS.

There were clear differences between the physics-constrained (MC-LSTM) and unconstrained (LSTM) data-driven models in the high-return period metrics. While both data-driven models performed better than both benchmark models in these out-

of-sample events, adding mass balance constraints resulted in *reduced* performance in the out-of-sample years.

The MC-LSTM includes a flux term that accounts for unobserved sources and sinks (e.g., evapotranspiration, sublimation, percolation). However, it is important to note that most or all hydrology models that are based on closure equations include a residual term in some form. Like all mass balance models, the MC-LSTM explicitly accounts for all water in and across the boundaries of the system. In the case of the MC-LSTM, this residual term is a single, aggregated flux that is parameterized with

weights that are *shared* across all 498 basins. Even with this strong constraint, the MC-LSTM performs significantly better than the physically-based benchmark models. This result indicates that classical hydrology model structures (conceptual flux equations) actually cause larger prediction errors than can be explained as being due to errors in the forcing and observation data.





**Table 2.** Median performance metrics across 498 basins on two separate time split test periods and test period split by return period (or probability) of the annual peak flow event (i.e., testing across years with an a peak annual event above 5 year return period, or a 20 percent probability of annual exceedance).

| Metric | Test period: 1989 - 1999 | | | Test period: 1996 - 2014 | | | | Test period: low probability years | | | |
| --- | --- | --- | --- | --- | --- | --- | --- | --- | --- | --- | --- |
| | LSTM | MC-LSTM | SAC-SMA | LSTM | MC-LSTM | SAC-SMA | NWM-Rv2 | LSTM | MC-LSTM | SAC-SMA | NWM-Rv2 |
| NSE | 0.72 | 0.71 | 0.64 | 0.71 | 0.72 | 0.63 | 0.63 | 0.81 | 0.77 | 0.66 | 0.67 |
| KGE | 0.73 | 0.73 | 0.67 | 0.77 | 0.74 | 0.68 | 0.67 | 0.77 | 0.71 | 0.62 | 0.64 |
| Pearson-r | 0.86 | 0.86 | 0.82 | 0.86 | 0.86 | 0.81 | 0.82 | 0.91 | 0.9 | 0.84 | 0.85 |
| Alpha-NSE | 0.82 | 0.82 | 0.79 | 0.94 | 0.87 | 0.83 | 0.85 | 0.82 | 0.77 | 0.7 | 0.79 |
| Beta-NSE | -0.04 | -0.02 | -0.01 | 0.01 | -0.01 | -0.01 | -0.01 | -0.03 | -0.04 | -0.03 | -0.04 |
| FHV | -17.95 | -16.76 | -19.74 | -7.17 | -13.1 | -15.55 | -13.02 | -17.37 | -24.08 | -31.08 | -20.42 |
| FLV | -8.37 | -33.74 | 31.18 | -9.49 | -27.23 | 28.56 | 2.85 | -2.49 | -39.39 | 27.1 | 10.81 |
| FMS | -7.28 | -8.79 | -14.27 | -9.67 | -8.65 | -8.38 | -5.23 | -6.37 | -4.87 | -11.29 | -4.31 |
| Peak-Timing | 0.33 | 0.33 | 0.43 | 0.38 | 0.4 | 0.53 | 0.54 | 0.36 | 0.42 | 0.72 | 0.62 |



## 3.2 Benchmarking Peak-Flows

Figure 1 shows the average absolute percent bias of annual peak flows for water years with different return periods. The training/calibration period for these results is the contiguous test period (water years 1996-2014). All models had increasingly large average errors with increasingly large extreme events. LSTM average error was lowest in all the return period bins. SAC-SMA was the worst performing model in terms of average error. SAC-SMA was trained (calibrated) on the same data as the LSTM and MC-LSTM, and its performance decreased substantively with increasing return period while that of the LSTM did

not.

Figure 2 shows the average absolute percent bias of annual peak flows for water years with different return periods, from models with train/test split based on return periods, with all test data coming from water years 1996-2014. This means that Figures 1 and 2 are only partially comparable – all statistics for each return period bin were calculated on the same observation data. All of the data shown in Figure 1 come from the test period. However since all water years with return periods of less

than 5 years were used for training in the return-period based train/test split, the 1-5 year return period category on Figure 2 shows metrics calculated on training data. What is comparable from these two figures are relative trends between models.

For the return-period test (Figure 2) the LSTM, MC-LSTM, and SAC-SMA were trained on data from all water years in 1980-2014 with return periods smaller or equal to 5 years, and all of the models showed substantively better average performance in the low return period (high probability) events than in the high return period (low probability) events. SAC-SMA

performance deteriorated faster than LSTM and MC-LSTM performance with increasingly extreme events. The unconstrained data-driven model (LSTM) performed better on average than all physics-informed and physically-based models in predicting extreme events in all out-of-sample training cases except for the 25-50 and 50-100, where the NWM-Rv2 performed slightly better on average. However, remember that the NWM-Rv2 calibration data was not segregated by return period.

## 4 Conclusions & Discussion

The hypothesis tested in this work was that data-driven streamflow models are likely to become unreliable in extreme or out-of-sample events. This is an important hypothesis to test because it is a common concern among physical scientists and among users of model-based information products (e.g., Todini, 2007), however prior work (e.g., Kratzert et al., 2019b; Gauch et al., 2021) demonstrated that data-based rainfall-runoff models were more reliable than other types of physically-based models, even in extrapolation to ungauged basins (Kratzert et al., 2019a). Our results indicate that this hypothesis is incorrect – the

data-driven models (both the pure ML model and the physics-informed ML model) were better than benchmark models at predicting peak flows in almost all conditions, including extreme events and including when extreme events were not included in the training data set.

It was somewhat surprising to us that the physics-constrained LSTM did not perform as well as the pure LSTM at simulating peak flows and out-of-sample events. This surprised us for two reasons. First, we expected that adding closure would help in

situations where the model sees rainfall events that are larger than anything it had seen during training. In this case, the LSTM could simply 'forget' water while the MC-LSTM would have to do something with the excess water – either store it in cell



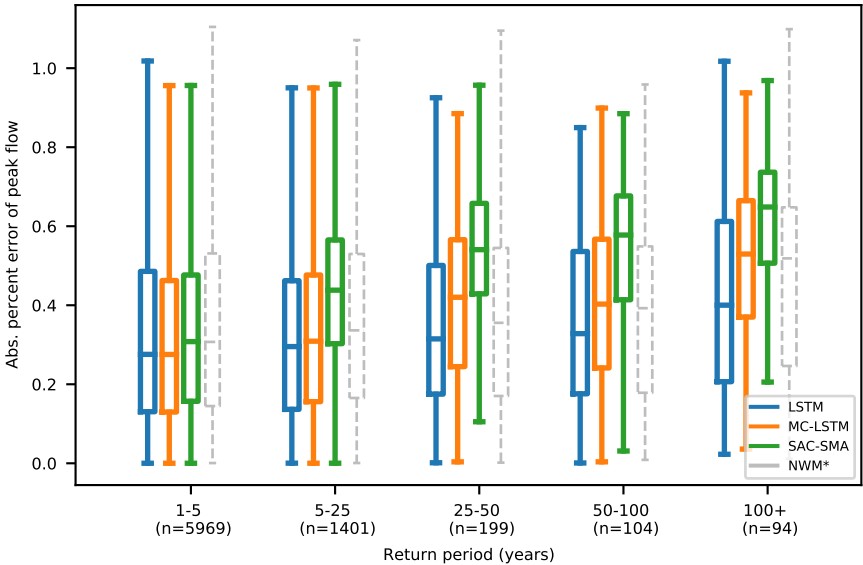

**Figure 1.** Average absolute percent bias of daily peak flow estimates from four models binned down by return period, showing results from models trained on a contiguous time period that contains a mix of different peak annual return periods. All statistics shown are calculated on test period data. The LSTM, MC-LSTM, and SAC-SMA models were all trained (calibrated) on the same data and time period. The NWM was calibrated on with the same forcing data, but on a different time period.

states or release it through one of the output fluxes. Second, Hoedt et al. (2021) reported that the MC-LSTM had lower bias than the LSTM on 98th percentile streamflow events (this is our FHV metric). Our comparison between different training/test periods showed that FHV is a volatile metric, which might account for this discrepancy. The analysis by Hoedt et al. (2021)

also did not consider whether a peak flow event was similar or dissimilar to training data, and we saw the greatest differences between the LSTM and MC-LSTM when predicting out-of-sample return period events.

This finding (differences between pure ML and physics-informed ML) is worth discussing. The project of adding physical constraints to ML is an active area of research across most fields of science and engineering (Karniadakis et al., 2021), including hydrology (e.g., Zhao et al., 2019; Jiang et al., 2020; Frame et al., 2020). It is important to understand that there is only one

type of situation in which adding any type of constraint (physically-based or otherwise) to a data-driven model can add value: if constraints help optimization. Helping optimization is meant here in a very general sense, which might include processes such as smoothing the loss surface, casting the optimization into a convex problem, restricting the search space, etc. Neural networks (and recurrent neural networks) can emulate large classes of functions (Hornik et al., 1989; Schäfer and Zimmermann, 2007), and by adding constraints to this type of model we can only *restrict* (not expand) the space of possible functions that

the network can emulate. This form of regularization is valuable *only* if it helps locate a better (in some general sense) local minimum on the optimization response surface (Mitchell, 1980). And it is *only* in this sense that that constraints imposed by physical theory can add information relative to what is available purely from data.

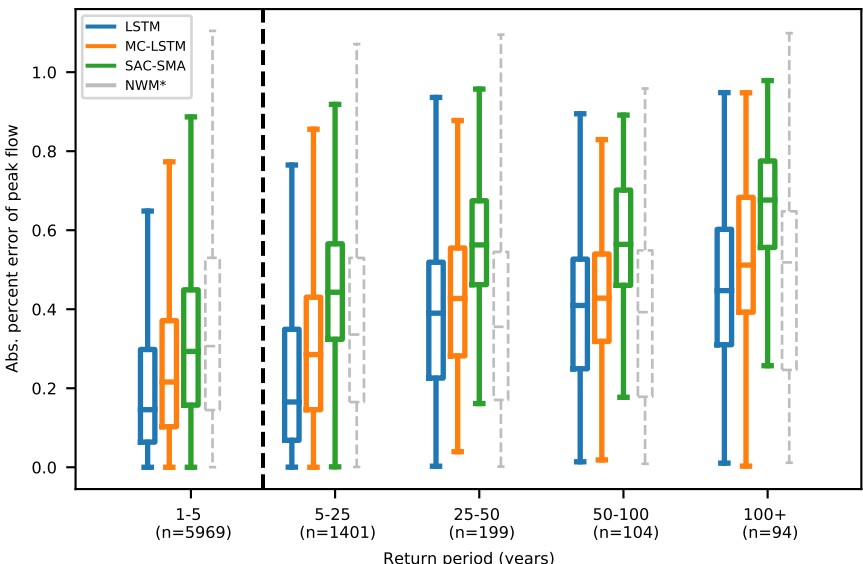

**Figure 2.** Average absolute percent bias of daily peak flow estimates from four models binned down by return period, showing results from models trained only on water years with return periods less than 5 years. The 1-5 year return period bin (left of the black dashed line) show statistics calculated on training data, while bins with return period years 5+ (to the right of the black dashed line) show statistics calculated on testing data. The LSTM, MC-LSTM, and SAC-SMA models were all trained (calibrated) on the same data and time period. The NWM was calibrated on with the same forcing data, but on a contiguous time period that does not exclude extreme events, as described in section 2.3.2

**Appendix A: LSTM**

Long Short Term Memory networks (Hochreiter and Schmidhuber, 1997) represent time-evolving systems using a recurrent
network structure with an explicit state space. Although LSTMs are not based on physical principles, Kratzert et al. (2018) argued that they are useful for rainfall-runoff modeling because they represent dynamic systems in a way that corresponds with physical intuition – specifically, LSTMs are Markovian in the (weak) sense that the future depends on the past only conditionally through the present state and future inputs. This type of temporal dynamics is implemented in an LSTM using an explicit input-state-output relationship that is conceptually similar to most hydrology models.
The LSTM architecture (Figure A1) takes a sequence of input features $\boldsymbol{x} = [\boldsymbol{x}[1], ..., \boldsymbol{x}[T]]$ of data over $T$ time steps, where each element $\boldsymbol{x}[t]$ is a vector containing features at time step $t$. A vector of recurrent *cell states* $\boldsymbol{c}$ is updated based on the input features and current cell state values at time $t$. The cell states also determine LSTM outputs or hidden states, $\boldsymbol{h}[t]$ , which are passed through a *head layer* that combines the LSTM outputs (that are not associated with any physical units) into predictions $\hat{\boldsymbol{y}}[t]$ that attempt to match the target data (which may or may not be associated with physical units).



The LSTM structure (without the head layer) is as follows:

$$\boldsymbol{i}[t] = \sigma(\boldsymbol{W_i}\boldsymbol{x}[t] + \boldsymbol{U_i}\boldsymbol{h}[t-1] + \boldsymbol{b_i}) \tag{A1}$$

$$\boldsymbol{f}[t] = \sigma(\boldsymbol{W_f}\boldsymbol{x}[t] + \boldsymbol{U_f}\boldsymbol{h}[t-1] + \boldsymbol{b_f}) \tag{A2}$$

$$\boldsymbol{g}[t] = \tanh(\boldsymbol{W_g}\boldsymbol{x}[t] + \boldsymbol{U_g}\boldsymbol{h}[t-1] + \boldsymbol{b_g}) \tag{A3}$$

$$\boldsymbol{o}[t] = \sigma(\boldsymbol{W_o}\boldsymbol{x}[t] + \boldsymbol{U_o}\boldsymbol{h}[t-1] + \boldsymbol{b_o}) \tag{A4}$$

$$\boldsymbol{c}[t] = \boldsymbol{f}[t] \odot \boldsymbol{c}[t-1] + \boldsymbol{i}[t] \odot \boldsymbol{g}[t] \tag{A5}$$

$$\boldsymbol{h}[t] = \boldsymbol{o}[t] \odot \tanh(\boldsymbol{c}[t]), \tag{A6}$$

The symbols $\boldsymbol{i}[t]$, $\boldsymbol{f}[t]$ and $\boldsymbol{o}[t]$ refer to the *input gate*, *forget gate*, and *output gate* of the LSTM respectively, $\boldsymbol{g}[t]$ is the *cell input* and $\boldsymbol{x}[t]$ is the *network input* at time step $t$, $\boldsymbol{h}[t-1]$ is the LSTM output, which is also called the *recurrent input* because it is used as inputs to all gates in the next timestep, and $\boldsymbol{c}[t-1]$ is the cell state from the previous time step.

Cell states represent the memory of the system through time, and are initialized as a vector of zeros. $\sigma(\cdot)$ are sigmoid activation functions, which return values in $[0,1]$. These sigmoid activation functions in the forget gate, input gate, and output gate are used in a way that is conceptually similar to on/off switches – multiplying anything by values in $[0,1]$ is a form of attenuation. The forget gate controls the memory timescales of each of the cell states, and the input and output gates control flows of information from the input features to the cell states and from the cell states to the outputs (recurrent inputs),

respectively. $\boldsymbol{W}$, $\boldsymbol{U}$ and $\boldsymbol{b}$ are calibrated parameters, where subscripts indicate which gate the particular parameter matrix/vector is associated with. $\tanh(\cdot)$ is the hyperbolic tangent activation function, which serves to add nonlinearity to the model in the cell input and recurrent input, and $\odot$ indicates element-wise multiplication. For a hydrological interpretation of the LSTM, see Kratzert et al. (2018).

## Appendix B:  Mass Conserving LSTM

The LSTM has an explicit input-state-output structure that is recurrent in time and is conceptually similar to how physical scientists often model dynamical systems. However the LSTM does not obey physical principles, and the internal cell states have no physical units. We can leverage this input-state-output structure to enforce mass conservation, in a manner that is similar to discrete-time explicit integration of a dynamical systems model, as follows:

$$\textit{New States} = \textit{Old States} + \textit{Inputs} - \textit{Outputs}. \tag{B1}$$

Using the notation from Appendix A, this is:

$$\boldsymbol{c}^*[t] = \boldsymbol{c}^*[t-1] + \boldsymbol{x}^*[t] - \boldsymbol{h}^*[t], \tag{B2}$$

where $\boldsymbol{c}^*[t]$, $\boldsymbol{x}^*[t]$ and $\boldsymbol{h}^*[t]$ are components of the cell states, input features, and model outputs (recurrent inputs) that contribute to a particular conservation law.




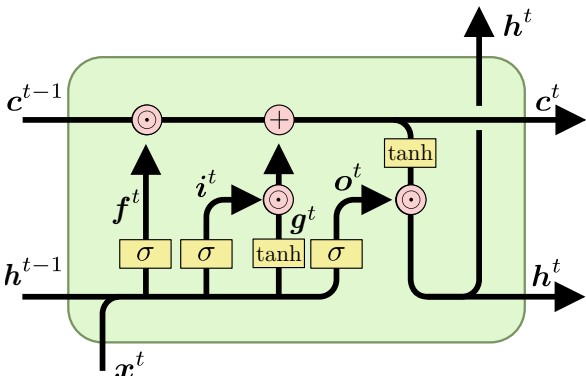

**Figure A1.** A single timestep of a standard LSTM with timesteps marked as superscripts for clarity. $\boldsymbol{x}^t$, $\boldsymbol{c}^t$, and $\boldsymbol{h}^t$ are the input features, cell states, and recurrent inputs at time $t$, respectively. $\boldsymbol{f}^t$, $\boldsymbol{i}^t$, and $\boldsymbol{o}^t$ are the forget-, input- and output-gate and $\boldsymbol{g}^t$ denotes the cell input. Boxes labeled $\sigma$ and tanh represent single sigmoid and hyperbolic tangent activation layers with the same number of nodes as cell states. The addition sign represent element-wise addition and $\odot$ represents element-wise multiplication.

As presented by Hoedt et al. (2021), we can enforce conservation in the LSTM by doing two things. First, we use special
activation functions in some of the gates to guarantee that mass is conserved from the inputs and previous cell states. Second,
we subtract the outgoing mass from the cell states. The important property of the special activation functions is that the sum of
all elements sum to one. This allows the outputs of each activation node to be scaled by a quantity that we want to conserve, so
that each scaled activation value represents a fraction of that conserved quantity. In practice, we can use any standard activation
function (e.g., sigmoid, ReLU), as long as we normalize the activation. With positive activation functions we can, for example,
normalize by the L1 norm (see Eq. B3 and B4). Another option would be to use the softmax activation function, which sums
to one by definition.

$$\widehat{\sigma}(s_k) = \frac{\sigma(s_k)}{\sum_k \sigma(s_k)} \tag{B3}$$

$$\widehat{\text{ReLU}}(s_k) = \frac{\max(s_k, 0)}{\sum_k \max(s_k, 0)} \tag{B4}$$

The constrained model architecture is illustrated in Fig. B1. An important difference with the standard architecture is that
the inputs are separated into *mass inputs* $\boldsymbol{x}$ and *auxiliary inputs* $a$. In our case, the mass input is precipitation and the auxiliary
inputs are everything else (e.g. temperature, radiation, catchment attributes). The input gate (sigmoids) and cell input (hyper-
bolic tangents) in the standard LSTM are (collectively) replaced by one of these normalization layers, while the output gate
is a standard sigmoid gate, similar to the standard LSTM. The forget gate is also replaced by a normalization layer, with the
important difference that the output of this layer is a square matrix with dimension equal to the size of the cell state. This matrix
is used to "reshuffle" the mass between the cell states at each timestep. This *reshuffling matrix* is column-wise normalized so
that the dot product with the cell state vector at time $t$ results in a new cell state vector having the same absolute norm (so that
no mass is lost or gained).





We call this general architecture a *Mass-Conserving LSTM* (MC-LSTM), even though it works for any type of conservation law (mass, energy, momentum, counts, etc.). The architecture is illustrated in Figure B1 and is described formally as follows:

$$\hat{\boldsymbol{c}}[t-1] = \frac{\boldsymbol{c}[t-1]}{||\boldsymbol{c}[t-1]||_1} \tag{B5}$$

$$\boldsymbol{i}[t] = \widehat{\sigma}(\boldsymbol{W}_i\boldsymbol{x}[t] + \boldsymbol{U}_i\hat{\boldsymbol{c}}[t-1] + \boldsymbol{V}_i\boldsymbol{a}[t] + \boldsymbol{b}_i) \tag{B6}$$

$$\boldsymbol{o}[t] = \sigma(\boldsymbol{W}_o\boldsymbol{x}[t] + \boldsymbol{U}_o\hat{\boldsymbol{c}}[t-1] + \boldsymbol{V}_o\boldsymbol{a}[t] + \boldsymbol{b}_o) \tag{B7}$$

$$\boldsymbol{R}[t] = \widehat{\mathrm{ReLU}}(\boldsymbol{W}_R\boldsymbol{x}[t] + \mathrm{U}_R\hat{\boldsymbol{c}}[t-1] + \mathrm{V}_R\boldsymbol{a}[t] + \boldsymbol{b}_R) \tag{B8}$$

$$\boldsymbol{m}[t] = \boldsymbol{R}[t]\boldsymbol{c}[t-1] + \boldsymbol{i}[t]\boldsymbol{x}[t] \tag{B9}$$

$$\boldsymbol{c}[t] = (1 - \boldsymbol{o}[t]) \odot \boldsymbol{m}[t] \tag{B10}$$

$$\boldsymbol{h}[t] = \boldsymbol{o}[t] \odot \boldsymbol{m}[t] \tag{B11}$$

Learned parameters are $\boldsymbol{W}$, $\boldsymbol{U}$, $\boldsymbol{V}$, and $\boldsymbol{b}$ for all of the gates. The normalized activation functions are, in this case, $\widehat{\sigma}$ (see Eq. B3) for the input gate and $\widehat{\mathrm{ReLU}}$ (see Eq. B4) for the redistribution matrix $\boldsymbol{R}$, as in the hydrology example of Hoedt et al. (2021). The product of $\boldsymbol{i}[t]\boldsymbol{x}[t]$ and $\boldsymbol{o}[t] \odot \boldsymbol{m}[t]$ are input and output fluxes, respectively.

Because this model structure is fundamentally conservative, all cell states and information transfers within the model are associated with physical units. Our objective in this study was to maintain the overall water balance in a catchment – our conserved input feature, $\boldsymbol{x}$, is precipitation in units $[mm/day]$ and our training targets are catchment discharge also in units of $[mm/day]$. Thus, all input fluxes, output fluxes, and cell states in the MC-LSTM have units of $[mm/day]$.

In reality, precipitation and streamflow are not the only fluxes of water into or out of a catchment. Because we did not provide
the model with (for example) observations of evapotranspiration, aquifer recharge, or baseflow, we accounted for unobserved sinks in the modeled systems by allowing the model to use one cell state as a *trash cell*. The output of this cell is ignored when we derive the final model prediction as the sum of the outgoing mass $\sum \boldsymbol{h}$.

## Appendix C: Benchmarking annual return period metrics

Figure C1 shows nine performance metrics calculated on model test results split into bins according to the return period of the
peak annual flow event. The LSTM, MC-LSTM and SAC-SMA were calibrated/trained on water years 1981-1995. The results shown in this figure are for water years 1996-2014. The LSTM and MC-LSTM performs better than the benchmark models according to most metrics, and during most return period bins. There are a few instances where the NWM performs better than the LSTM and/or the MC-LSTM. The NWM calibration does not correspond to the training/calibration period of SAC-SMA, LSTM or the MC-LSTM.

Figure C2 shows the nine performance metrics calculated on model test results split into bins according to the return period of the peak annual flow event. The LSTM, MC-LSTM and SAC-SMA were calibrated/trained on water years with a peak annual flow event that had a return period of less that five years (i.e., bin 1-5 indicated by the dashed line). The results shown in this figure are for water years 1996-2014. The LSTM and MC-LSTM performs better than the SAC-SMA model according every





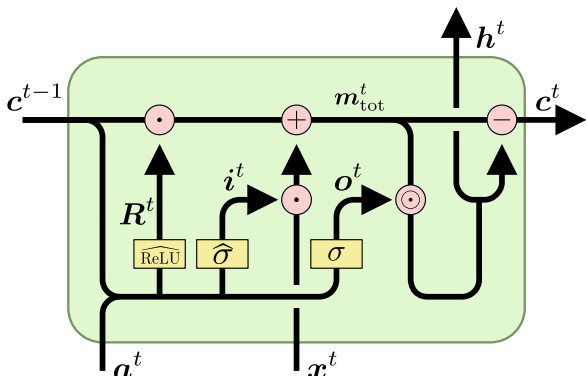

**Figure B1.** A single timestep of a Mass-Conserving LSTM with timesteps marked as superscripts for clarity. As in Figure A1, $c^t$, $a^t$, $x^t$, $i^t$, $o^t$, and $R^t$ are the cell states, conserved inputs, input features, input fluxes, output fluxes, and reshuffling matrix at time $t$, respectively. $\sigma$ represents a standard sigmoid activation layer, $\widehat{\sigma}$ and $\widehat{\mathrm{ReLU}}$ represent normalized sigmoid activation layers and normalized ReLU activation layer respectively. Addition and subtraction signs represent element-wise addition and subtraction, $\odot$ represents element-wise multiplication and the $\cdot$ sign represents the dot-product.

metric, and during all bins. There are a few instances where the NWM performs better than the LSTM and/or the MC-LSTM.

The NWM calibration does not correspond to the training/calibration period of SAC-SMA, LSTM or the MC-LSTM.



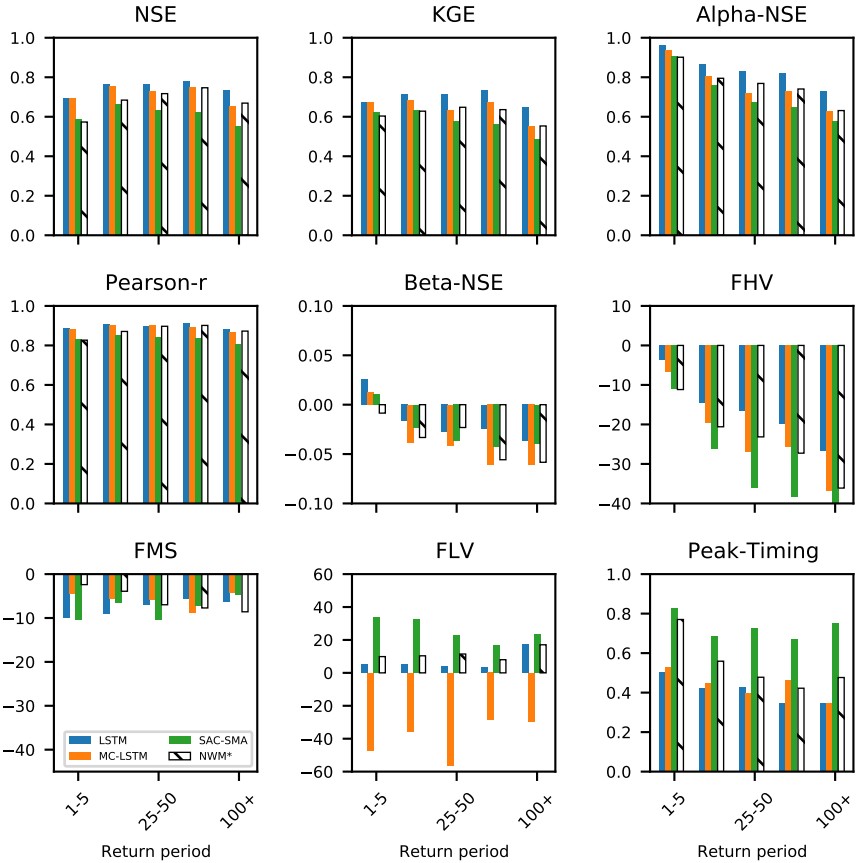

**Figure C1.** Metrics for training only on a standard time split; train period was water years 1981-1995 and test period (shown here) was water years 1996-2014.

*Code and data availability.* All LSTMs and MC-LSTMs were trained using the NeuralHydrology Python library available at https://github.com/neuralhydrology/neuralhydrology. A snapshot of the exact version that we used is available at https://github.com/jmframe/mclstm_2021_extrapolate/neuralhydrology and under DOI number 10.5281/zenodo.5051961. Code for calibrating SAC-SMA is from https://github.com/Upstream-Tech/SACSMA-SNOW17, which includes the SpotPy calibration library https://pypi.org/project/spotpy/. Input data for all
model runs except the NWM-Rv2 came from the public NCAR CAMLES repository https://ral.ucar.edu/solutions/products/camels and were used according to instructions outlined in the NeuralHydrology readme. NWM-Rv2 data are available publicly from https://registry.opendata.aws/nwm-archive/. Code for the return period calculations is publicly available from https://www.mathworks.com/matlabcentral/fileexchange/22628-log-pearson-flood-flow-frequency-using-usgs-17b (Burkey, 2009), and daily USGS peak flow data extracted from the USGS Water Information System for the CAMELS return period analysis were collected and archived on the CUAHSI HydroShare platform
under DOI number 10.4211/hs.c7739f47e2ca4a92989ec34b7a2e78dd. All model output data generated by this project will be available on the CUAHSI HydroShare platform under a DOI number (after revisions). Interactive Python scripts for all post-hoc analysis reported in this paper,





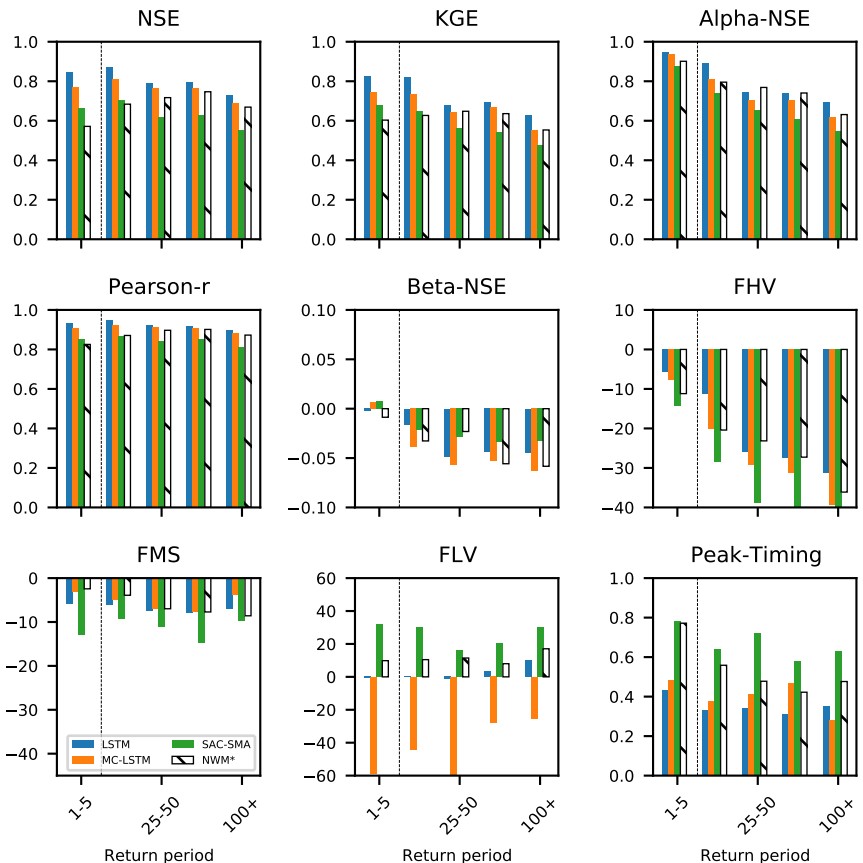

**Figure C2.** Metrics for the models trained only on high-probability years. The bins of return periods greater than 5 are out-of-sample for the LSTM, MS-LSTM and SAC-SMA.

including calculating metrics and generating tables and figures, are available at https://github.com/jmframe/mclstm_2021_extrapolate/results and under DOI number 10.5281/zenodo.5165216

.



*Author contributions.* Jonathan Frame conceived the experimental design, contributed to the manuscript and performed experiments and analysis. Frederik Kratzert, Daniel Klotz, and Martin Gauch wrote the LSTM and MC-LSTM code, as well as all code for metrics calculations, and participated in analysis and interpretation of results. Oren Gilion and Hoshin Gupta participated in interpretation of results. Logan Qualls assisted with data prepossessing. Grey Nearing advised on experimental design, helped setup training, calibration, and model runs except NWM-Rv2, wrote the manuscript and supervised the research project.

*Competing interests.* The authors report no competing interests.

*Acknowledgements.* Jonathan Frame and Grey Nearing were partially supported by a grant from the NASA Terrestrial Hydrology Program (award #80NSSC18K0982). Frederik Kratzert was supported by a Google Faculty Research Award (PI: Sepp Hochreiter). We further acknowledge support by Verbund AG for Daniel Klotz and by the Linz Institute of Technology DeepFlood project for Martin Gauch.





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
