# Peer review of "Deep learning rainfall-runoff predictions of extreme events"

_Hydrology and Earth System Sciences, 2021_

## Referee Comment (RC3)

**Review: Deep learning rainfall-runoff predictions of extreme events**

This paper seeks to answer an extremely pertinent question, to what extent do LSTMs (the current state of the art for rainfall-runoff modelling) continue to perform adequately when predicting out of sample extreme events? This idea that LSTMs will fail whenever there are extreme events that we do not observe in the training period has been a key criticism of data-driven approaches and yet there has been very little empirical validation of this. This paper is well timed and addresses this research gap.

The main results are drawn from the experiment which separates years of data by the peak flow in that year, thus simulating unseen extremes, and therefore testing model adequacy in a future with more extreme flows.

Ultimately, this is an extremely valuable contribution to HESS and I recommend that it is published.

**Comments:**

The paper is incredibly clear, with a clear hypothesis: "*data-driven streamflow models are likely to become unreliable in extreme or out- of-sample events*" and a clear conclusion: "reject the hypothesis". I really enjoyed the separation of text and appendices. It makes the material much more easily digestible and allows extra information to be kept for the interested reader.

This is potentially out of scope and I do not want to add clutter to this paper since it is incredibly concise and a valuable contribution. That being said, I am left wondering how these patterns vary spatially over the 498 basins used for training. Is there anything that can be said about the model conditions in which the LSTM/MC-LSTM/SAC-SMA difference is large? Is it the same as found by Nearing et al (2020) where the post-processed SAC-SMA model was most improved in snowy catchments? Please feel free to ignore this recommendation!

Is there any way that we can see a hydrograph showing the difference for an extreme high-flow in the two experiments: 1) showing how well the models perform when the training period is split the same way as the original papers cited 2) showing how the performance differs when the training period is split by the max annual return period. This may also not be necessary if the authors feel that a single example will focus too much attention on a sample of 1. I understand that choosing an archetypal example can be difficult and potentially lead to cherry-picking of results. Completely up to the authors!

L112-129: This paragraph describes the main experimental setup for the paper. Why do the authors use a threshold of a 5-year return period (20% chance of annual exceedance) split for

train-test? Was it because it allowed a sufficient number of samples in the train/test split? Would it be possible to make this choice explicit?

Figure C2: Looking at the peak-timing metric below, it seems as though the MC LSTM (**orange**) outperforms the LSTM on high return period flows. I recognise that the performance seems at odds with the other results, but is one interpretation of this finding that Mass-Conservation helps with peak-timing for the really high return period flows?

[Figure]

Figure C2: I find the following graph interesting, and perhaps worthy of discussion (at least in the appendix C). The results for the MC-LSTM / LSTM seem to be most different for the FLV metric (looking at the bias of the low-flows). What is it about the physical constraints that could cause such large differences for the low flows? Is it in these low-flow conditions that errors in the underlying data (and therefore the mass-balance) have the largest impact?

[Figure]

**Minor Comments:**

L44-45: "*the basins that were used for model benchmarking by Newman et al. (2017), who removed basins with (i) large discrepancies between different methods of calculating catchment area, and (ii) areas larger than 2,000 km2.*" What is the justification for removing these larger catchment areas?

L47: "because we benchmarked against the National Water Model retrospective" → "because we benchmarked against the National Water Model retrospective**ly**" ? I'm not sure what is trying to be said here?

L298-299: "*In our case, the mass input is precipitation and the auxiliary inputs are everything else (e.g. temperature, radiation, catchment attributes)*" Is there any way that energy can be conserved using this model too? In terms of the energy available for evaporation of water? This is definitely out of scope. I am just interested in whether this is possible given the approach.

L320-322 "*we accounted for unobserved sinks in the modeled systems by allowing the model to use one cell state as a trash cell. The output of this cell is ignored when we derive the final model prediction as the sum of the outgoing mass*" This is incredibly interesting! Is there any information about the times/locations when this "trash cell" contains a lot of water? Not for this paper but if you had any sense of whether these learned outflows correspond with subsurface transfers of water then that would be super interesting!

L323-325: "The NWM calibration does not correspond to the training/calibration period of SAC-SMA, LSTM or the MC-LSTM." Can we possibly have a more immediate answer to what time period was the NWM calibration? It's written elsewhere () but since this is an appendix it would help the reader to know what the NWM calibration was here too.
"*was calibrated by NOAA personnel on about 1400 basins with NLDAS forcing data on water years 2009--2013*"

L333: "SAC-SMA model according every" → "SAC-SMA model according **to** every"

**References**

Nearing, G., Sampson, A. K., Kratzert, F., and Frame, J.: Post-processing a Conceptual Rainfall-runoff Model with an LSTM, 2020a.

---

## Author Response (AR1)

Dear Nadav Peleg,

Thank you for the opportunity to submit a revised manuscript for our work "Deep learning rainfall-runoff predictions of extreme events". Below are detailed replies (blue text) to all of the referees' comments (grey text). Also included in the replies are quotations of changes to the manuscript (red text). We do include some figures in these replies for the purpose of responding to the reviewers' comments, and these figures are also included in the open discussion on Copernicus, but we do not intend to add these figures to the manuscript, as they are not suitable for publication.

Please let us know if you have any questions.

Jonathan M. Frame

**Replies to RC1: Anonymous Referee #1, 05 Sep 2021**

Anonymous Referee #1 provided very thoughtful comments on our paper. They ask for additional commentary on a few specific topics. After careful consideration of the reviewers comments, we believe that our original manuscript covers these topics satisfactorily.

This is an interesting paper that addresses a most relevant issue of incorporating background knowledge about processes in question into machine learning (ML) algorithms.

We are glad you find the paper interesting, thank you.

Clearly, in ML we are hoping to program computers by telling them what we want to achieve without having to explicitly instruct them how to achieve such goals. But, what is it that we want from those programs? Do they just need to be accurate or should we also be able to interpret them?

What we want from hydrology models depends on the application but certain applications (such as flood forecasting and water management) require the most accurate models possible.

As we mention at the end of Appendix A, for a hydrological interpretation of the LSTM, see Kratzert et al. (2018).

This idea that ML models are not interpretable is out-of-date. ML theory has most or all of the same basic building blocks that physical theory has (Lin et al., 2017). This "understanding" does not always look like how a traditional hydrologist or traditional physicist views the problem, but it is no less descriptive, robust, or interpretable. When someone claims that ML models are uninterpretable, this, frankly, tells us mostly that the person expressing the sentiment has chosen not to do this, instead of saying anything about the models (or ML theory) themselves. The mechanisms represented by data-driven models are interpretable in any meaningful sense, so long as the user is willing to do so (Nearing et al., 2020, Shwartz-Ziv and Tishby, 2017).

In the scientific contexts, the ambition is clear: we are looking for a learning machine capable of finding an accurate approximation of a natural phenomenon, as well as expressing it in the form of a meaningful or an interpretable model. The authors adopt this view, and I fully subscribe to such a working hypothesis.

At the same time, this bias towards meaningfulness and interpretability opens several additional issues. The computer-generated hypotheses should take advantage of the already existing body of knowledge about the domain in question. In the case of two equally good approximations of the same data set: the one which blindly fits the data and the other which, in addition to the fit, also respects the background knowledge, we should be biased toward the latter one.

The hypothetical scenario you present here is not applicable to this paper. In all serious benchmarks conducted in (surface) hydrology (streamflow, surface fluxes, snow, etc.), all traditional models are significantly worse than ML models in all cases. This hypothetical equivalence simply does not exist within the discipline in any meaningful way.

Perhaps more importantly, in this paper in particular (as described in Section 2.3 Models), our paper presents four models 1) a "conceptual" model in SAC-SMA, 2) a "process-based" model in NWM-Rv2, 3) a "pure" ML model in the LSTM, and 4) a "physics-informed" ML model in the MC-LSTM. Our results show that these models do not have equally good approximations of the same data set, and our conclusion states that "the data-driven models (both the pure ML model and the physics-informed ML model) were better than benchmark models at predicting peak flows in almost all conditions, including extreme events and including when extreme events were not included in the training data set." We further conclude that "the physics-constrained LSTM did not perform as well as the pure LSTM at simulating peak flows and out-of-sample events." It is my opinion that these results show that the pure LSTM respects the background knowledge (the data collected of observable hydrologic conditions, in this case streamflow) better than the other three models.

We reject the characterization that ML models "blindly" fit data. There is a large amount of background knowledge included in these models, just not in the way that this reviewer is used to thinking about systems. The difference between these models and traditional hydrology models (that embed traditional ways of understanding catchments) is that these models actually perform well. We have to ask, given how (universally) bad traditional hydrology models are, do they really encode background knowledge, or maybe just background fantasies and speculation? We invite the reviewer to take some time to learn how physical understanding and prior knowledge is encoded into deep learning models, which is relatively standard practice in (modern) ML, across disciplines.

However, there are few questions that I would appreciate authors could address in the manuscript to a greater depth.

1.) The fashion in which we express knowledge about the processes and make it available to the learning machine remains rather unclear.

The conceptual bias of the LSTM for this application is that the LSTM, as any traditional hydrology model, is a state-space model. The model processes the inputs sequentially and updates internal states (similar to hyd models). The difference is that the state representations and update functions are learned from data and are not hard coded. This is explained in Appendix A: LSTM & Appendix B: Mass Conserving LSTM.

One can insist on strict adherence to the background knowledge principles - such as 100% mass balance accuracy. We declare this desire and hence this is referred to as declarative bias.

100% mass balance accuracy is not possible in watershed models due to unobservable mass inputs and losses (such as inter-watershed subsurface flow and anthropogenic water resources). Also, hydrology models usually have unobserved terms, such as evaporation or deep percolation which break the observable mass balance.

Alternatively, one can treat the bias as an additional objective that should be treated simultaneously with the goodness of fit in the learning process. This is referred to as preferential bias. Declarative bias reduces search space but results in so-called broken ergodicity.

Preferential bias results in a Pareto-optimal set of solutions. For further discussion in the context of water management see:

M Keijzer and V Babovic, 2002, Declarative and preferential bias in GP-based scientific discovery, Genetic Programming and Evolvable Machines 3 (1), 41-79

And according to Keijzer and Babovic, reducing the search space does not help finding better solutions faster. We address this in our paper on lines 236-242 of the original manuscript. "if constraints help optimization. Helping optimization is meant here in a very general sense, which might include processes such as smoothing the loss surface, casting the optimization into a convex problem, restricting the search space, etc. Neural networks (and recurrent neural networks) can emulate large classes of functions (Hornik et al., 1989; Schäfer and Zimmermann, 2007), and by adding constraints to this type of model we can only restrict (not expand) the space of possible functions that the network can emulate. This form of regularization is valuable only if it helps locate a better (in some general sense) local minimum on the optimization response surface (Mitchell, 1980). And it is only in this sense that constraints imposed by physical theory can add information relative to what is available purely from data."

Also preferential bias has the same problem. It's only pareto-optimal on the space of allowed functions. This is true for any model trained with declarative bias too. A model that allows for conceptual flexibility does so via some (well-defined mathematical structure - any training procedure is constrained by that structure. It does not matter if we feel like we might learn something "physical" (or otherwise) after training. Searching for a larger class of functions is always better, except when it isn't.

In the present paper, it would appear that authors prefer declarative treatment of background knowledge. However, I would appreciate further analysis, comparison, and, if that is not possible, at least a discussion on preferential vs. declarative bias in the case studies described in theirs work.

No, we do not prefer declarative treatment of background knowledge. According to Keijzer and Babovic, reducing the search space "does not help finding better solutions faster. In fact, for the class of scientific discovery problems the opposite seems to be the case." We come to the same conclusion in our paper (lines 234-242): "It is important to understand that there is only one type of situation in which adding any type of constraint (physically-based or otherwise) to a data-driven model can add value: if constraints help optimization. Helping optimization is meant here in a very general sense, which might include processes such as smoothing the loss surface, casting the optimization into a convex problem, restricting the search space, etc. Neural networks (and recurrent neural networks) can emulate large classes of functions (Hornik et al., 1989; Schäfer and Zimmermann, 2007), and by adding constraints to this type of model we can only restrict (not expand) the space of possible functions that the network can emulate. This form of regularization is valuable only if it helps locate a better (in some general sense) local minimum on the optimization response surface (Mitchell, 1980). And it is only in this sense that constraints imposed by physical theory can add information relative to what is available purely from data." We do not believe that further discussion is required.

2.) Bias Variance Tradeoff. Arguably incorporation of the knowledge bias affects model variance. In this case, bias denotes the difference between the average prediction of a model and the correct value which it is trying to predict. Variance is the variability of model prediction for a given data point or a value that tells us the spread of our data. For in-depth discussion see:

Hastie, T; Tibshirani, R; Friedman, J. H. (2009). The Elements of Statistical Learning, Springer

I would love to see a more in-depth analysis of the bias-variance tradeoff in the present case, and am looking forward to reading more about it in the revised version of the manuscript.

This is well covered in our paper. We train/calibrate our models using the Nash-Suttcliffe Efficiency (NSE), which when decomposed includes a term for bias (Ratio of the means of observed and simulated flow) and a term for variance (Ratio of standard deviations of observed and simulated flow). There is certainly a bias-variance tradeoff in our trained/calibrated models, but the NSE as a loss function is a good way to include these two terms. In Table 2 we present the results of both the Apha-NSE (The variance term), and the Beta-NSE (The bias term).

According to Elements of Statistical Learning "to trade bias off with variance in such a way as to minimize the test error." The bias-variance tradeoff is an analysis to see how a model generalizes to be used on data that is not part of the training set. We show in our results that the LSTM model generalizes from a training set without extremely large runoff events to low probability, high flow events, that are not included in the training set.

High bias can cause an algorithm to miss the relevant relations between features and target outputs. High variance may result from an algorithm modeling the random noise in the training data

3.) Models vs. Predictions.

LSTM-type of ML models are extremely good at forecasting. The authors have eloquently argued in favour of the approach in this (as well as in previous) published research works. At the same time, one must consider if such an ML approaches induce models or forecasters. On this topic I would advise the following recent works:

HMVV Herath, et al, 2021, Hydrologically informed machine learning for rainfall–runoff modelling: towards distributed modelling, Hydrology and Earth System Sciences 25 (8), 4373-4401

HMVV Herath, J et al, 2021, Genetic programming for hydrological applications: to model or forecast that is the question, Journal of Hydroinformatics

**"In our public and private sector work with operational ML models, we have found that these models are very conducive to operational work by traditional hydrologists. We do different things to make the models "accessible" to different audiences, but that is out-of-scope for this paper.**

In general, this is an interesting and potentially valuable contribution to the hydrological society. I am looking forward to reading revised version of the manuscript.

In general, the reviewer seems to be making philosophical arguments about machine learning, and we fail to see almost any relevance in these arguments to what we are doing in this paper. We appreciate the reviewer's comment about bias-variance tradeoff (as this is at least directly relevant to the paper), however the reviewer seems to have missed that this is treated explicitly already.

**Citation: https://doi.org/10.5194/hess-2021-423-RC1**

Lin, Henry W., Max Tegmark, and David Rolnick. "Why Does Deep and Cheap Learning Work So Well?" *Journal of Statistical Physics* 168, no. 6 (2017): 1223–47. https://doi.org/10.1007/s10955-017-1836-5.

Nearing, Grey S., Frederik Kratzert, Alden Keefe Sampson, Craig S. Pelissier, Daniel Klotz, Jonathan M. Frame, Cristina Prieto, and Hoshin V. Gupta. "What Role Does Hydrological Science Play in the Age of Machine Learning?" *Water Resources Research*, 2020. https://doi.org/10.1029/2020wr028091.

Shwartz-Ziv, Ravid, and Naftali Tishby. "Opening the Black Box of Deep Neural Networks via Information," 2017, 1–19. http://arxiv.org/abs/1703.00810.

**Replies to RC2: Anonymous Referee #2, 23 Sep 2021**

Anonymous Referee #2 pointed out an anomaly in our results (with the FLV metric) that requires some additional explanation. Below we include a few figures that may be useful to the review of the changes made to the new manuscript, but we do not believe that they are suitable to include in the manuscript itself.

Recently, deep learning has d more accurate rainfall-runoff predictions than a conceptual and physically-based model; this is to be expected as they are trained to be accurate predictively. It is also interesting that these models can generalise to different catchments and are trying to explain some of the mechanisms leading to runoff.

The paper provides a first-hand comparison of how different models (deep learning vs conceptual vs physical model) fare on unseen extreme events. It is clear that with the comparison, the deep learning-based model outperforms on different accuracy metrics.

The paper also argues that deep learning provides the hydrological sciences community's most accurate rainfall-runoff simulations. While this might be true, but certainly requires comparison with many available different models in different parts of the globe. Moreover, this might be only true when we have a large amount of data available. Nevertheless, I would agree that deep learning provides one of the most accurate rainfall-runoff predictions. In the future, there is much potential for a deep learning-based model for rainfall-runoff prediction.

One of the vital points raised in the paper is about the conceptual flux equations and highlights the potential for improvement with a comparison to MC-LSTM.

Another exciting thing is about the FLV (Bottom 30% low flow bias). Analysing why FLV increases(in magnitude) drastically for MC-LSTM could be an interesting direction to explore. Especially for the low probability years. As theoretically, the machine learning model should have seen such low flow data. The author illustrates that any constraint restricts the space of possible functions that the network can emulate. MC-LSTM is developed primarily to model this type of situation where an entity is conserved. Furthermore, unlike other metrics, which did not deteriorate much, we see a drastic drop (increase in magnitude) in FLV. More analysis in this direction would be interesting for the readers as well.

There is a large discrepancy of the MC-LSTM for the FLV metric. It turns out that outliers play a major role in the FLV results. And that they are regionally clustered, mostly, around the south-west. Basically, because the FLV depends on a log value of the simulation and observed flows, they have to be either removed, or artificially set above zero. This causes a very large instability in the calculation. Basically, this metric is not viable when streamflow approaches zero. Below is a little further analysis. I believe we can convey the following in the discussion, without adding additional figures.

**FLV, LSTM - MC-LSTM**

Flow duration curve (left) and flow duration curve of the minimum 30% of flows for Basin 09513780, which is in the cluster of Arizona basins shown on the map above. The actual curves of the LSTM and the MC-LSTM are not that far off, but the difference in FLV metric between the two is wildly different (~1e10) : lstm -9.506222e+11, mc-lstm -1.907157e+11, sac -1.517753e+11.

We will add the following discussion of the FLV metric describing the fragile, and potentially misleading results:.

"The results indicate that the MC-LSTM performs much worse according to the FLV metric, but we caution that the FLV metric is fragile, particularly when flows approach zero (due to dry or

frozen conditions). The large discrepancy comes from several outlier basins that are regionally clustered, mostly, around the south-west. The FLV equation includes a log value of the simulation and observed flows. This causes a very large instability in the calculation. Flow duration curves (and flow duration curve of the minimum 30% of flows) of the LSTM and the MC-LSTM are qualitatively similar, but they diverge on the low flow in terms of log values."

As mentioned above, the FLV has some problems when flows approach zero, so we will add the description above. Thank you for pointing this anomaly out.

The paper highlights the potential of deep learning models to predict extreme events, while the hypothesis is that the data-driven models lose reliability in extreme events more than models based on process-understanding. The notion of reliability can be somewhat vague and should be clarified. The paper is only focusing on the predictive reliability here.

Will clarify about focusing on "predictive reliability".

Abstract: [Line 2] "There is a concern among hydrologists that the predictive accuracy of data-driven models based on deep learning may not be reliable in extrapolation or for predicting extreme events."

Introduction: [Line 14] "Echoing this sentiment about the perceived predictive reliability of data-driven models" & [Line 17] "The idea that the predictive accuracy of hydrological models based on physical understanding might be more reliable than machine learning (ML) based models in out-of-sample conditions was drawn from early experiments on shallow neural networks" & [Line 29] "The primary objective of this study is to test the hypothesis that data-driven models lose predictive accuracy in extreme events more than models based on process-understanding."

Methods: [Line 72] "testing whether the predictive accuracy of the LSTM is reliable in extreme events" &

Conclusion & Discussion: [Line 221] "The hypothesis tested in this work was that predictions made by data-driven streamflow models are likely to become unreliable in extreme or out-of-sample events." & [Line 224] "prior work \citep[e.g.,]]]{kratzert2019bench, gauch2021rainfall} demonstrated that predictions made by data-based rainfall-runoff models were more reliable than other types of physically-based models"

Overall, the paper would provide the first comparison on predictive accuracy for unseen extreme events for a deep learning model and a valuable contribution to the hydrological community.

Citation: https://doi.org/10.5194/hess-2021-423-RC2

**Replies to RC3: Anonymous Referee #3, 26 Nov 2021**

Anonymous Referee #3 provided very positive, but constructive feedback on our manuscript. Below we address the reviewer's questions, which includes some additional figures and results that we do not think are necessary to be included in the manuscript.

RC3: ["I am left wondering how these patterns vary spatially over the 498 basins used for training."]

An obvious geospatial pattern would be very satisfying for these results. Unfortunately, it does not appear that a strong geospatial pattern exists, so we decided to leave any such results out of the manuscript. Other publications include the spatial maps of the SAC-SMA, NWM and LSTM results. Below is the spatial pattern of the DIFFERENCE between LSTM and MC-LSTM. Some small clusters exist where the difference of LSTM and MC-LSTM is positive or negative, but nothing particularly exciting. It does appear that there is larger magnitude difference between LSTM and MC-LSTM in the middle longitudes of CONUS. However, this is due to the fact that all models tend to perform poorly in this region. If the differences were consistently positive (LSTM performs better than MC-LSTM) or consistently negative (MC-LSTM performs better than LSTM), then that would be an interesting results. But unfortunately it appears that it is mostly random where the LSTM and MC-LSTM perform differently. NLDAS forcings tend to create larger discrepancies between the two ML models than Daymet forcings, but that is because Daymet forcings are more informative (Kratzert et al, 2020), so both the LSTM and MC-LSTM make better predictions with Daymet.

**NSE, LSTM - MC-LSTM**

RC3: ["Is there anything that can be said about the model conditions in which the LSTM/MC-LSTM/SAC-SMA difference is large?"]

**There are no obvious general conditions that lead to large discrepancies. Perhaps it might be possible to make inferences about the conditions at individual basins.**

RC3: ["Is it the same as found by Nearing et al (2020) where the post-processed SAC-SMA model was most improved in snowy catchments?"]

**We also do not see that in this data here.**

RC3: ["Is there any way that we can see a hydrograph showing the difference for an extreme high-flow in the two experiments: 1) showing how well the models perform when the training period is split the same way as the original papers cited 2) showing how the performance differs when the training period is split by the max annual return period. This may also not be necessary if the authors feel that a single example will focus too much attention on a sample of 1. I understand that choosing an archetypal example can be difficult and potentially lead to cherry-picking of results. "]

Just as you caution, I am very reluctant to cherry pick some hydrographs. Perhaps we can host the complete series of hydrographs on Hydroshare.

RC3: ["L112-129: This paragraph describes the main experimental setup for the paper. "]

RC3: ["Why do the authors use a threshold of a 5-year return period (20% chance of annual exceedance) split for train-test? "]

This was an arbitrary choice made before the experiment was run. Perhaps we could have done lower to get more "extreme" high flow events, as there are plenty of lower flow years for training, but I would be worried about potential "p-hacking" type of activity.

RC3: ["Was it because it allowed a sufficient number of samples in the train/test split? Would it be possible to make this choice explicit? "]

We did consider the number of training/testing splits, but just in the sense that once we picked 5-yr as the threshold, we checked to make sure we had enough data for both training and testing. Again, we picked this threshold BEFORE looking at the segregated data or conducting any experiments, in order to avoid experimental bias. RC3: ["Figure C2: Looking at the peak-timing metric below, it seems as though the MC LSTM (orange) outperforms the LSTM on high return period flows. I recognise that the performance seems at odds with the other results, but is one interpretation of this finding that Mass-Conservation helps with peak-timing for the really high return period flows? "]

That is a possibility, but given the relatively low sample size of the 100+ year events (84) as compared to the events above 5-yr and below 100-yr (1536), that result is not particularly robust.

We've added the sample size for the results in Appendix C: "The total number of samples in each bin are as follows: n=5969 for 1-5, n=1260 for 5025, n=185 for 25-50, n=91 for 50-100 and n=84 for 100+."

RC3: ["Figure C2: I find the following graph interesting, and perhaps worthy of discussion (at least in the appendix C). The results for the MC-LSTM / LSTM seem to be most different for the FLV metric (looking at the bias of the low-flows). What is it about the physical constraints that could cause such large differences for the low flows? Is it in these low-flow conditions that errors in the underlying data (and therefore the mass-balance) have the largest impact? "]

Our response to Reviewer 2 provides figures that show the FLV metric is wildly unstable when flows are very low, or zero. We will add the following discussion on this metric:

"The results indicate that the MC-LSTM performs much worse according to the FLV metric, but we caution that the FLV metric is fragile, particularly when flows approach zero (due to dry or frozen conditions). The large discrepancy comes from several outlier basins that are regionally clustered, mostly, around the south-west. The FLV equation includes a log value of the simulation and observed flows. This causes a very large instability in the calculation. Flow duration curves (and flow duration curve of the minimum 30% of flows) of the LSTM and the MC-LSTM are qualitatively similar, but they diverge on the low flow in terms of log values."

RC3: ["Minor Comments: "]

RC3: ["L44-45: "the basins that were used for model benchmarking by Newman et al. (2017), who removed basins with (i) large discrepancies between different methods of calculating catchment area, and (ii) areas larger than 2,000 km2." What is the justification for removing these larger catchment areas? "]

Critically, the reason that the current study removes these basins is to be consistent with previous benchmarking experiments, specifically the now rather large set of community experiments that

inherit from Newman 2017. Andy Newman's justification for doing this initially back in 2017 was that the basin area is used to convert between streamflow [L3/T] and surface runoff depth [L]. Since the two basin area calculations disagree by a large magnitude, we cannot be sure which is closer to the truth.

RC3: ["L47: "because we benchmarked against the National Water Model retrospective"  $\rightarrow$  "because we benchmarked against the National Water Model retrospectively" ? I'm not sure what is trying to be said here? "]

The National Water Model Retrospective is a specific dataset. The word "retrospective" in the name of this dataset refers to the fact that these data are model hindcasts. We make this more clear by changing the phrasing to include the new documentation "NOAA National Water Model CONUS Retrospective Dataset". A link to these data is included in the data availability statement.

RC3: ["L298-299: "In our case, the mass input is precipitation and the auxiliary inputs are everything else (e.g. temperature, radiation, catchment attributes)" Is there any way that energy can be conserved using this model too? In terms of the energy available for evaporation of water? This is definitely out of scope. I am just interested in whether this is possible given the approach. "]

The basic structure of the MC-LSTM is conservative – any quantity or quantities can be conserved (e.g., mass, energy, momentum, counts, etc.). In the current study, we aren't using surface flux data, so it would not be possible to use the CAMELS data to (reasonably) constrain energy. But we have done this with FluxNet data in other studies.

RC3: ["L320-322 "we accounted for unobserved sinks in the modeled systems by allowing the model to use one cell state as a trash cell. The output of this cell is ignored when we derive the final model prediction as the sum of the outgoing mass" This is incredibly interesting! Is there any information about the times/locations when this "trash cell" contains a lot of water? Not for this paper but if you had any sense of whether these learned outflows correspond with subsurface transfers of water then that would be super interesting! "]

**That is planned for another experiment, possibly even in review already.**

RC3: ["L323-325: "The NWM calibration does not correspond to the training/calibration period of SAC-SMA, LSTM or the MC-LSTM." Can we possibly have a more immediate answer to what time period was the NWM calibration?"]

**Yes. Good idea. Added "The NWM calibration does not correspond to the training/calibration period of SAC-SMA, LSTM or the MC-LSTM." to Appendix C.**

RC3: ["It's written elsewhere () but since this is an appendix it would help the reader to know what the NWM calibration was here too. "was calibrated by NOAA personnel on about 1400 basins with NLDAS forcing data on water years 2009--2013""]

RC3: ["L333: "SAC-SMA model according every" → "SAC-SMA model according to every""]

Thank you, it has been revised. "SAC-SMA model according to every"